# Metabolic Engineering for Efficient Synthesis of Patchoulol in *Saccharomyces cerevisiae*

**Qiu Tao** [1,2,3,4], **Guocheng Du** [1,2,3,4], **Jian Chen** [1,2,3,4], **Juan Zhang** [1,2,3,4,*] and **Zheng Peng** [1,2,3,4,*]

1   Key Laboratory of Industrial Biotechnology, Ministry of Education, School of Biotechnology, Jiangnan University, 1800 Lihu Road, Wuxi 214122, China; 6210208053@stu.jiangnan.edu.cn (Q.T.); gcdu@jiangnan.edu.cn (G.D.); jchen@jiangnan.edu.cn (J.C.)
2   Science Center for Future Foods, Jiangnan University, 1800 Lihu Road, Wuxi 214122, China
3   Engineering Research Center of Ministry of Education on Food Synthetic Biotechnology, Jiangnan University, 1800 Lihu Road, Wuxi 214122, China
4   Jiangsu Province Engineering Research Center of Food Synthetic Biotechnology, Jiangnan University, 1800 Lihu Road, Wuxi 214122, China
*   Correspondence: zhangj@jiangnan.edu.cn (J.Z.); zhengpeng@jiangnan.edu.cn (Z.P.)

**Abstract:** Patchoulol is a natural sesquiterpene alcohol with extensive applications in cosmetics and pharmaceuticals. In this study, we first constructed the synthesis pathway of patchoulol in *Saccharomyces cerevisiae* by expressing the patchoulol synthase *PTS* gene using the strong promoter *GAL1*. Afterward, the metabolic flux of the precursor was enhanced by strengthening the mevalonate pathway and balancing the precursor competition pathway, resulting in a 32.74-fold increase in patchoulol production. Subsequently, the supply of acetyl-CoA in yeast was increased by modifying transcriptional regulators and modulating the acetyl-CoA pathway, and the titer of patchoulol reached 155.94 mg/L. Finally, optimization of the fermentation conditions resulted in a titer of 195.96 mg/L in the shake flasks. Further, batch-fed fermentation in a 5 L bioreactor yielded 1.95 g/L. This work accelerated the development of a microbial cell factory for the production of patchoulol.

**Keywords:** patchoulol; *Saccharomyces cerevisiae*; metabolic engineering; mevalonate pathway; Acetyl-CoA

## 1. Introduction

Patchoulol, a sesquiterpene alcohol present in *Pogostemon cabin* leaves, is widely used in cosmetics, skin care products and perfumes because of its unique aroma and lasting fragrance [1,2]. In addition, patchoulol has extensive pharmacological effects, such as neuroprotection, resistance to inflammation, anticancer, analgesia, etc. Patchoulol is also used in the synthesis of the chemotherapy drug Taxol, so it has great potential value in the field of medicine [3]. Due to the low and unstable annual production of patchoulol, the price of patchoulol fluctuates between $30–200/kg [4]. At present, patchoulol is mainly extracted from *Pogostemon cabin* by steam distillation. However, this method consumes a lot of electric energy and kerosene and is affected by plant growth, planting location and climate environment, so the product quality is unstable and the cost is high [5,6]. Compared with the plant extraction method, biosynthesis has the advantages of not being restricted by the environment, short cycle time and simple production operation, which is a very promising method that can be used for large-scale production.

Among existing engineered microorganisms, *S. cerevisiae* is extensively used in the production of terpenoids owing to its safety, high catabolic rate of glucose, clear genetic background and relatively fast growth rate [7]. *S. cerevisiae* cells contain the mevalonate (MVA) pathway, which can produce the common precursors of terpenoids, isopentenyl pyrophosphate (IPP) and dimethylallyl pyrophosphate (DMAPP). IPP and DMAPP are converted to farnesyl pyrophosphate (FPP) under the catalysis of endogenous farnesyl pyrophosphate synthase (*FPPS*), which is a direct precursor of patchoulol and converted

to patchoulol under the catalysis of exogenous patchoulol synthase (*PTS*). In recent years, researchers have attempted to synthesize patchoulol using *S. cerevisiae* and have made significant progress. Asadollahi et al. [8] constructed a pathway for the synthesis of patchoulol in *S. cerevisiae*, and reduced the metabolic flow of squalene synthesis by regulating the *ERG9* gene, so that the titer of patchoulol reached 16.9 mg/L. Ma et al. [9] achieved a patchoulol yield of 466.8 mg/L in a 5 L fermenter by fusing *FPPS* with *PTS*, enhancing the MVA pathway, and inhibiting the synthesis of farnesol and ergosterol. Liu et al. [6] modified the metabolic pathway of yeast, enhanced the expression of the ergosterol pathway, modified the synthase of patchoulol, and then obtained 1632 mg/L patchoulol by fed-batch fermentation. Despite the fact that the yield of patchoulol has been greatly improved, it still cannot meet the existing demand.

The study focused on enhancing patchoulol production in *S. cerevisiae* through combinatorial metabolic engineering (Figure 1). Initially, the gene *PTS* was heterogeneously expressed using the inducible promoter *GAL1* to generate a patchoulol-producing strain. Subsequently, it was divided into four modules for systematic metabolic regulation of yeast strains, which involved strengthening the endogenous MVA synthesis pathway, reducing competition from the precursor FPP pathway, eliminating endogenous transcriptional regulatory factors and enhancing the acetyl-CoA pathway. The performance of the recombinant strains was assessed based on cell growth, squalene content and patchoulol yield. Through optimization of fermentation conditions and implementation of fed-batch fermentation, the patchoulol yield in *S. cerevisiae* was significantly increased.

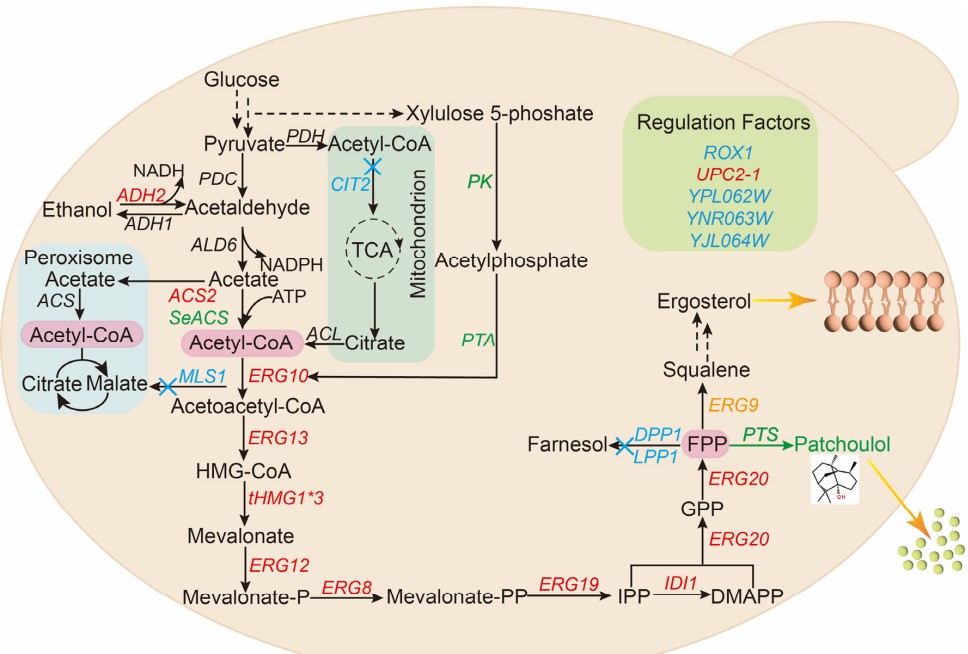

**Figure 1.** Schematic diagram of combining metabolic engineering to increase the production of patchoulol in *S. cerevisiae*. The red fonts indicate enhanced genes; the green fonts indicate the introduced foreign gene; the blue fonts indicate the knocked-out gene; and the yellow fonts indicate the weakened gene. Dashed arrows indicate multi-step reactions. *ADH2*, alcohol dehydrogenase; *ACS2*, acetyl-CoA synthetase; *ERG10*, acetoacetyl-CoA thiolase; *ERG13*, hydroxymethylglutaryl-CoA synthase; *tHMG1*, catalytic domain of hydroxymethylglutaryl-CoA reductase; *ERG12*, mevalonate kinase; *ERG8*, phosphomevalonate kinase; *ERG19*, mevalonate pyrophosphate decarboxylase; *IDI1*, isopentenyl diphosphate isomerase; *ERG20*, farnesyl pyrophosphate synthase; *PK*, phosphotransferase; *PTA*, phosphotransacetylase; *SeACS*, acetyl-CoA synthetase from *Salmonella enterica*; *PTS*, patchoulol synthase; *CIT2*, citrate synthase; *MLS1*, malate synthase; *DPP1*, diacylglycerol Pyrophosphate Phosphatase; *LPP1*, lipid phosphate phosphatase; and *ERG9*, squalene synthase.

## 2. Materials and Methods

### 2.1. Strains and Media

In this study, *S. cerevisiae* CEN.PK2-1C was used as the starting strain and cultivated at 30 °C with YPD medium. Nutritional deficiency medium SD/-Trp (20 g/L glucose, 6.7 g/L YNB, 80 mg/L his, 80 mg/L ura and 400 mg/L leu) and SD/Trp-Ura (SD/-Trp lacking ura) were used to screen and culture yeast inverters and SD/-Trp+5-FOA solid medium was used to reverse select URA3-labeled sgRNA expression plasmid [10]. *E. coli* DH5$\alpha$ was used for plasmid construction and cultured in LB medium supplemented with 100 mg/L ampicillin at 37 °C.

Prime STAR Max DNA polymerase was purchased from TaKaRa (Otsu, Japan). Plasmid extraction and DNA gel purification kits were purchased from Thermo Scientific (Wilmington, NC, USA). The standard of patchoulol, squalene and dodecane were purchased from Aladdin (Shanghai, China). YPD Broth was purchased from Haibo Biotechnology Co., LTD (Qingdao, China).

### 2.2. Plasmids Construction

All plasmids constructed in this study are listed in Table 1. The sequencing verification of DNA and the synthesis of primers were conducted by Sangon Biotech (Shanghai, China). The original plasmids pESC-URA, p426-SNR52p-gRNA and p414-TEF1p-Cas9-CYC1t were obtained from our laboratory store. The exogenous coding gene *PTS* was synthesized by GENEWIZ (Suzhou, China).

**Table 1.** Plasmids used in this study.

| Names | Description | Reference |
|---|---|---|
| pESC-URA | 2 μ, URA3, Amp | This Lab |
| p414-TEF1p-Cas9-CYC1t | *Cas9* expression vector | This Lab |
| p426-SNR52p-gRNA.CAN1.Y-SUP4t | gRNA expression vector | This Lab |
| YEplac181 | 2 μ, LEU2 | This Lab |
| YEplac181-PTS | P$_{TEF1}$-*PTS*-T$_{ADH1}$ cassette in YEplac181 | GENEWIZ |
| pESC-PTS | P$_{GAL1}$-*PTS*-T$_{CYC1}$ cassette in pESC-URA | This work |
| p426-YMRWΔ15 | gRNA expressing vector carrying *YMRWΔ15* crRNA sequence | This work |
| p426-GAL80 | gRNA expressing vector carrying *GAL80* crRNA sequence | This work |
| p426-HO | gRNA expressing vector carrying *HO* crRNA sequence | This work |
| p426-PDC6 | gRNA expressing vector carrying *PDC6* crRNA sequence | This work |
| p426-YORWΔ17 | gRNA expressing vector carrying *YORWΔ17* crRNA sequence | This work |
| p426-YPRCτ3 | gRNA expressing vector carrying *YPRCτ3* crRNA sequence | This work |
| p426-YNRCΔ9 | gRNA expressing vector carrying *YNRCΔ9* crRNA sequence | This work |
| p426-YERCΔ8 | gRNA expressing vector carrying *YERCΔ8* crRNA sequence | This work |
| p426-DPP1 | gRNA expressing vector carrying *DPP1* crRNA sequence | This work |
| p426-LPP1 | gRNA expressing vector carrying *LPP1* crRNA sequence | This work |
| p426-YPL062W | gRNA expressing vector carrying *YPL062W* crRNA sequence | This work |
| p426-YNR063W | gRNA expressing vector carrying *YNR063W* crRNA sequence | This work |
| p426-ROX1 | gRNA expressing vector carrying *ROX1* crRNA sequence | This work |
| p426 -YJL064W | gRNA expressing vector carrying *YJL064W* crRNA sequence | This work |
| p426-CIT2 | gRNA expressing vector carrying *CIT2* crRNA sequence | This work |
| p426-MLS1 | gRNA expressing vector carrying *MLS1* crRNA sequence | This work |
| p426-XI-3 | gRNA expressing vector carrying *XI-3* crRNA sequence | This work |
| p426-YORWΔ22 | gRNA expressing vector carrying *YORWΔ22* crRNA sequence | This work |
| p426-YPRCΔ15 | gRNA expressing vector carrying *YPRCΔ15* crRNA sequence | This work |
| p426-NDT80 | gRNA expressing vector carrying *NDT80* crRNA sequence | This work |
| p426-X-2 | gRNA expressing vector carrying *X-2* crRNA sequence | This work |
| p426-YHRCΔ14 | gRNA expressing vector carrying *YHRCΔ14* crRNA sequence | This work |
| p426-XI-2 | gRNA expressing vector carrying *XI-2* crRNA sequence | This work |
| p426-X-4 | gRNA expressing vector carrying *X-4* crRNA sequence | This work |
| p426-XI-1 | gRNA expressing vector carrying *XI-1* crRNA sequence | This work |
| p426-YARCΔ8 | gRNA expressing vector carrying *YARCΔ8* crRNA sequence | This work |

Construction of the pESC-PTS protein expression plasmid: the empty plasmid pESC-URA was transformed into a linear form with the primer pESC-F/R. Subsequently, the *PTS* fragment was amplified from *E. coli* DH5$\alpha$-PTS using the primer PTS-F/R. The two fragments were then assembled via OE-PCR and introduced into *E. coli* DH5$\alpha$. Verification of successful construction of the expression vector was conducted through colony PCR reactions and gene sequencing after single bacteria growth on an LB plate.

Construction of p426-sgRNA plasmid: crRNA sequence can be designed through an online website (CHOPCHOP (uib.no) (accessed 14 March 2023)) or independently based on CDS sequence. Primers were designed following the introduction of the crRNA sequence into the gRNA expression box. The full-length gRNA plasmid was amplified using a polymerase chain reaction, and the resulting product was then transformed into *E. coli* DH5$\alpha$ receptor cells to construct p426-sgRNA plasmid.

All of the primers and exogenous gene sequences used in this study are listed in Table S1 and Table S2, respectively.

### 2.3. Yeast Transformation and Strains Construction

This study utilized *CRISPR/Cas9* gene editing technology to construct yeast strains. Initially, the *Cas9* expression plasmid p414-TEF1p-Cas9-CYC1t was introduced into CEN.PK2-1C for subsequent gene editing. All strains constructed in this study are detailed in Table 2. Upstream and downstream homologous arms (approximately 500 bp), along with promoters, terminators and endogenous coding genes were amplified from the genomic DNA of *S. cerevisiae*. These components were then assembled into a fragment using OE-PCR to form either a gene knockout cassette or a gene expression cassette. The sgRNA plasmid and gene expression cassette were transferred into *S. cerevisiae* using the lithium acetate conversion method, and transformants were screened on SD/-Trp-Ura plates. Successfully transformed strains were cultivated on SD/-Trp+5-FOA plates to eliminate the sgRNA plasmid tagged with URA3 for subsequent transformations. The sgRNA plasmid-deleted strains were preserved in the SD/-Trp medium. The schematic representation of the strain construction process in this study is depicted in Figure S2.

**Table 2.** Strains used in this study.

| Names | Description | Reference |
|---|---|---|
| *S. cerevisiae* CEN.PK2-1C | Mata; his3$\Delta$1; leu2-3-112; ura3-52; trp1-289; MAL2-8c; SUC2 | This Lab |
| P414 | CEN.PK2-1C, contains p414-TEF1p-Cas9-CYC1t plasmid | This work |
| Q01 | CEN.PK2-1C, *YMRW$\Delta$15*:: P$_{TEF1}$-PTS-T$_{ADH1}$ | This work |
| Q02 | CEN.PK2-1C, *YMRW$\Delta$15*:: P$_{GAL1}$-PTS-T$_{ADH1}$ | This work |
| Q03 | CEN.PK2-1C, *YMRW$\Delta$15*:: P$_{GAL7}$-PTS-T$_{ADH1}$ | This work |
| Q04 | CEN.PK2-1C, *YMRW$\Delta$15*:: P$_{CCW12}$-PTS-T$_{ADH1}$ | This work |
| Q05 | CEN.PK2-1C, *YMRW$\Delta$15*:: P$_{PGK1}$-PTS-T$_{ADH1}$ | This work |
| Q06 | CEN.PK2-1C, *YMRW$\Delta$15*:: P$_{TDH3}$-PTS-T$_{ADH1}$ | This work |
| Q07 | Q02, $\Delta$*GAL80* | This work |
| TQP01 | Q07, pESC-PTS | This work |
| TQP02 | TQP01, $\Delta$*HO*:: P$_{GAL7}$–tHMG1-T$_{CYC1}$ | This work |
| TQP03 | TQP02, $\Delta$*PDC6*:: T$_{GPM1}$-ERG20-P$_{PGK1}$-P$_{TEF1}$-IDI1-T$_{ACT1}$ | This work |
| TQP04 | TQP03, *YORW$\Delta$17*:: P$_{GPD1}$-ERG19-T$_{PGK1}$-P$_{CCW12}$-ERG10-T$_{ADH1}$-P$_{TEF1}$-ERG8-T$_{CYC1}$ | This work |
| TQP05 | TQP04, $\Delta$*YPRC$\tau$3*:: P$_{TDH3}$-ERG12-T$_{PGK1}$-P$_{CCW12}$-ERG13-T$_{ADH1}$ | This work |
| TQP06 | TQP05, *YNRC$\Delta$9*:: P$_{TDH3}$–tHMG1-T$_{CYC1}$ | This work |
| TQP07 | TQP06, YERC$\Delta$8:: P$_{GAL1}$–tHMG1-T$_{CYC1}$ | This work |
| TQP08 | TQP07, $\Delta$*DPP1* | This work |
| TQP09 | TQP07, $\Delta$*LPP1* | This work |
| TQP10 | TQP08, $\Delta$*LPP1* | This work |
| TQP11 | TQP10, P$_{ERG9}$→P$_{HXT1}$ | This work |
| TQP12 | TQP10, P$_{ERG9}$→P$_{ERG1}$ | This work |
| TQP13 | TQP12, $\Delta$*YPL062W* | This work |

**Table 2.** *Cont.*

| Names | Description | Reference |
|---|---|---|
| TQP14 | TQP12, $\Delta YNR063W$ | This work |
| TQP15 | TQP12, $\Delta ROX1$ | This work |
| TQP16 | TQP13, $\Delta YNR063W$ | This work |
| TQP17 | TQP16, $\Delta YJL064W$ | This work |
| TQP18 | TQP17, $\Delta ROX1$ | This work |
| TQP19 | TQP17, $\Delta ROX1::$ $P_{GAL1}$–$UPC2$-1-$T_{CYC1}$ | This work |
| TQP20 | TQP17, $\Delta CIT2$ | This work |
| TQP21 | TQP17, $\Delta MLS1$ | This work |
| TQP22 | TQP17, $\Delta XI$-3:: $P_{TDH3}$-$ADH2$-$T_{ADH1}$ | This work |
| TQP23 | TQP17, $YORW\Delta22::$ $P_{TDH3}$-$ACS2$-$T_{ADH1}$ | This work |
| TQP24 | TQP17, $YPRC\Delta15::$ $P_{TEF1}$-$SeACS$-$T_{CYC1}$ | This work |
| TQP25 | TQP17, $\Delta NDT80::$ $P_{GAL1}$-$PK$-$T_{PGK1}$–$T_{CYC1}$-$PTA$-$P_{TDH3}$ | This work |
| Q22 | TQP22, discard the PESC-PTS plasmid | This work |
| TQ01 | Q22, $\Delta X$-2:: $P_{GAL1}$-$PTS$-$T_{CYC1}$ | This work |
| TQ02 | TQ01, $YHRC\Delta14::$ $P_{GAL1}$-$PTS$-$T_{CYC1}$ | This work |
| TQ03 | TQ02, $\Delta XI$-2:: $P_{GAL1}$-$PTS$-$T_{CYC1}$ | This work |
| TQ04 | TQ03, $\Delta X$-4:: $P_{GAL1}$-$PTS$-$T_{CYC1}$ | This work |
| TQ05 | TQ04, $YPRC\Delta15::P_{GAL1}$-$PTS$-$T_{CYC1}$ | This work |
| TQ06 | TQ05, $YORW\Delta22::$ $P_{GAL1}$-$PTS$-$T_{CYC1}$ | This work |
| TQ07 | TQ06, $\Delta NDT80::$ $P_{GAL1}$-$PTS$-$T_{CYC1}$ | This work |
| TQ08 | TQ07, $\Delta XI$-1:: $P_{GAL1}$-$PTS$-$T_{CYC1}$ | This work |
| TQ09 | TQ08, $YARC\Delta8::$ $P_{GAL1}$-$PTS$-$T_{CYC1}$ | This work |
| *E. coli* DH5$\alpha$ | The strain for plasmid construction | This Lab |

*2.4. Cultivation in Shaking Flask*

A single colony of *S. cerevisiae* was inoculated into 50 mL culture tubes containing 5 mL YPD medium and cultured at 30 °C and 220 rpm for 18 h to obtain the primary seed solution. The solution was then transferred to 250 mL shake flasks containing 50 mL YPD medium at an inoculum of 1% and incubated under the same conditions. After 12 h, 10% filtered dodecane was added to initiate a two-phase fermentation lasting 120 h. At the end of the fermentation, the $OD_{600}$ of the fermentation broth and the yield of patchoulol in the dodecane phase were measured.

*2.5. Extraction and Detection Methods*

To quantify patchoulol, the fermentation liquid was collected in a centrifuge tube and centrifuged at 8000 rpm for 5 min. The upper organic phase was then collected, an appropriate amount of anhydrous sodium sulfate was added, and the mixture was gently inverted several times before being left for approximately 30 min for dewatering. The upper organic phase was centrifuged at 12,000 rpm for 2 min, diluted with n-hexane, and filtered using an organic filter membrane of 0.22 μm [9,11]. Patchoulol was quantitatively determined using GC-MS (GC-2010 Plus and GCMS-QP2010, Shimadzu, Kyoto, Japan) with DB-5MS (30 m × 0.25 mm, 0.25 μm, Agilent, Santa Clara, CA, USA). The initial oven temperature was 50 °C for 1 min, then raised to 200 °C and 280 °C at rates of 10 °C/min and 20 °C/min, respectively, and held at 280 °C for 9 min. 1 μL sample was injected in a non-shunt manner, with helium as the carrier gas. The scanning range was set at 50–300 *m/z*, and selected ion monitoring (SIM) was employed for quantitative analysis, with mass charge ratios (*m/z*) of 138, 161 and 222 [2,6,12].

In order to detect the yield of squalene, 500 μL of the fermentation liquid was transferred to a 2 mL fragmentation tube, centrifuged at high speed for 10 min, the supernatant was removed, and the cell precipitation was mixed with an equal volume of grinding beads (0.5 mm), followed by the addition of 1.5 mL acetone. Yeast cell lysis was performed using Fast Prep of MP Biomedicals (Irvine, CA, USA), resulting in squalene being present in the upper organic phase [13,14]. Squalene was then detected by GC-MS, with the initial chamber temperature set at 90 °C for 1 min, followed by an increase to 280 °C at 20 °C/min,

and a 15 min hold. The sample was injected in a non-shunt manner, and the scanning range was 35–500 $m/z$ [15].

For the determination of cell dry weight (CDW), a 1.5 mL EP tube was dried in a 105 °C oven and weighed on an electronic balance, with the initial weight of the EP tube recorded. Following centrifugation of 1 mL fermentation broth at 10,000 rpm for 5 min, the supernatant was discarded, and the bacterial cells were washed with sterile water, dried until no further change in weight, and then weighed and recorded as the final weight. The difference between the final and initial weights was calculated, with three parallel experiments conducted for each group [16]. The relationship between cell dry weight and $OD_{600}$ was determined as y = 0.2971x + 0.7756 (where y represents cell concentration in g DCW/L and x represents $OD_{600}$) (Figure S1).

To determine the glucose and ethanol content in the fermentation solution, 1 mL of the solution was centrifuged at high speed for 2–3 min, the supernatant was diluted appropriately, filtered and tested using a biosensor analyzer (Shenzhen Hillman Technology Co., LTD, Shenzhen, China).

*2.6. Fed-Batch Fermentation*

In a fed-batch fermentation process using a 5-L bioreactor (Diebold Bioengineering Co., Ltd., Shanghai, China) with a liquid volume of 2.4 L, the primary and secondary seed solutions were cultured in a YPD medium with an inoculation amount of 4%. The fed-batch fermentation utilized an OYPD medium with an inoculation amount of 8%. The fermentation conditions were controlled with a temperature of 30 °C, stirring speed ranging from 300 to 800 rpm, dissolved oxygen (DO) maintained below 40% and pH between 5.5 and 6.0. After 20 h of fermentation, 270 mL of dodecane was added.

Supplementary medium I for the first stage (1 L): 800 g glucose, 300 g yeast extract.
Supplementary medium II for the second stage (1 L): 60% ethanol.
Metal ion solution A (1 L): 5.75 g $ZnSO_4 \cdot 7H_2O$, 0.32 g $MnC1_2 \cdot 4H_2O$, 0.47 g $CoC1_2 \cdot 6H_2O$, 0.48 g $Na_2MoO_4 \cdot 2H_2O$, 2.9 g $CaC1_2 \cdot 2H_2O$, 2.8 g $FeSO_4 \cdot 7H_2O$, 80 mL 0.5 M EDTA, Adjust pH = 8.0.
Vitamin solution B (1 L): 0.05 g biotin, 1 g calcium pantothenate, l g nicotinic acid, 25 g myo-inositol, 1 g thiamine HCl, 1 g pyridoxal HCl, 0.02 g p-aminobenzoic acid.

**3. Results and Discussion**

*3.1. Construction of Synthetic Pathway of Patchoulol in Saccharomyces cerevisiae*

*S. cerevisiae* possesses a mevalonate pathway utilizing the intermediate metabolite acetyl-CoA as the precursor, which can synthesize the direct precursor FPP of patchoulol, and is a promising microorganism for the synthesis of patchoulol [9]. However, due to the lack of the essential gene for patchoulol synthesis in *S. cerevisiae*, external introduction of patchoulol synthase is necessary to enable patchoulol production [6]. In this study, patchoulol synthase (*PTS*) from *Pogostemon cablin* was selected for codon optimization and gene synthesis. *PTS* was heterologously expressed in the P414 strain using a strong promoter *TEF1* (Figure 2a), resulting in the generation of the initial strain Q01. Following fermentation for 120 h, GC-MS analysis detected 0.46 ± 0.01 mg/L (0.04 ± 0.001 mg/g DCW) of patchoulol in the recombinant strain Q01, while no patchoulol was detected in the control strain (Figure 2b,c), indicating successful construction of a patchoulol synthesis pathway in strain Q01.

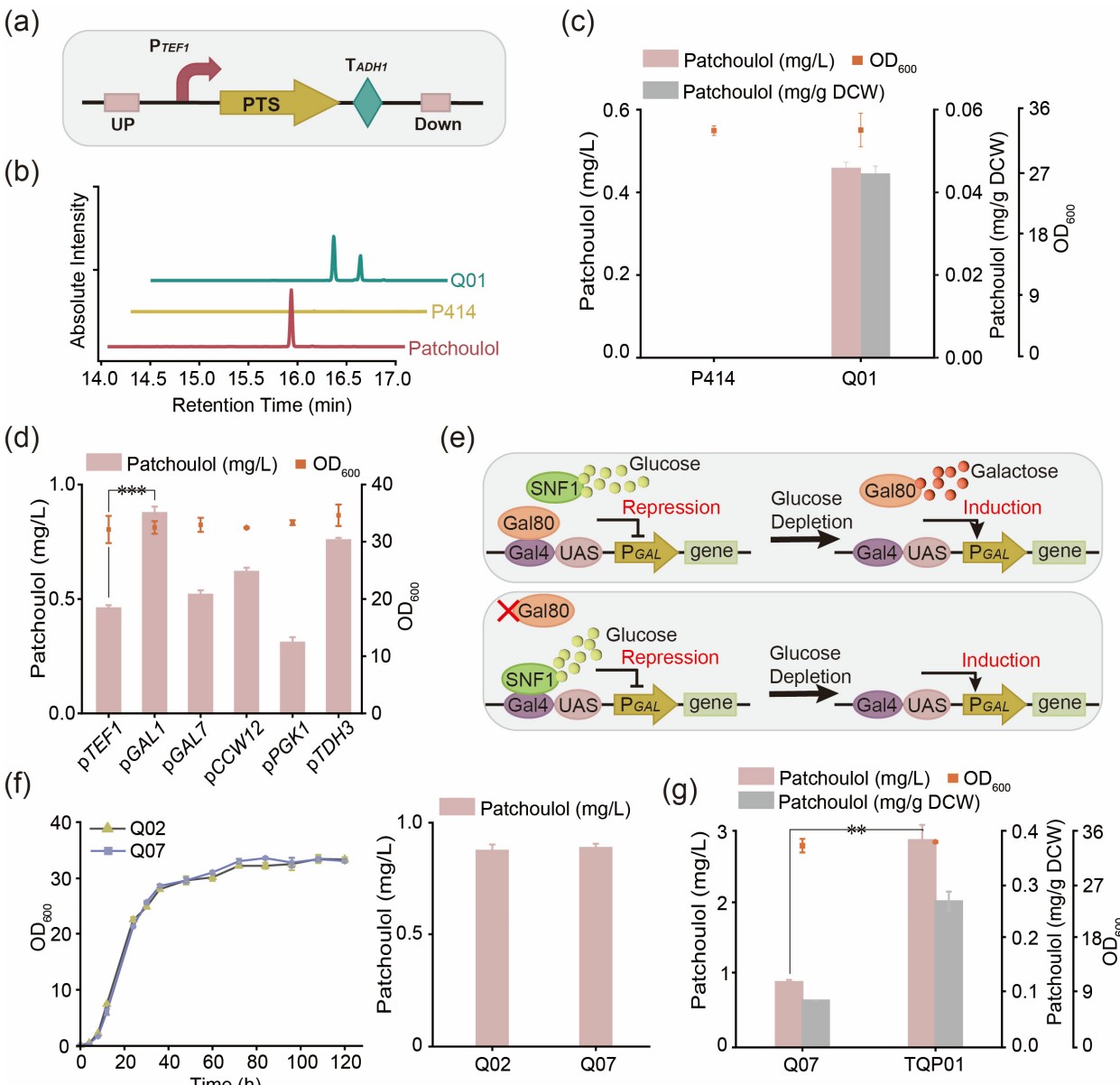

**Figure 2.** Construct the synthetic pathway of patchoulol in *S. cerevisiae.* (**a**) Schematic diagram of the gene expression box of patchoulol synthase; (**b**) GC-MS chromatograms of the standard of patchoulol and the engineered strain products. The peak time of patchoulol produced by strain Q01 was consistent with that of patchoulol standard; (**c**) Production of patchoulol from P414 and Q01 strains; (**d**) The growth and yield of *PTS* expressed by different promoters in shaker; (**e**) Knocking out *GAL80* removes the dependence of the inducible promoter on galactose. Red "X" indicates knockout; (**f**) The growth and yield of strains after knocking out *GAL80*; (**g**) The changes of yield after introduction of high copy plasmid pESC-PTS. Data presented as mean values ± SD from three independent biological replicates (*n* = 3). Statistical evaluation (*p*-value) compared to the control was conducted by a two-tailed *t*-test. ** $p < 0.01$, *** $p < 0.001$ ($p \geq 0.05$).

## 3.2. Promoter Screening for Rational Control of Patchoulol Biosynthesis

Promoters play a crucial role in microbial genetic systems by influencing gene transcription levels and ultimately impacting the production of metabolites [17]. In this study, the endogenous strong promoters *GAL1*, *GAL10*, *CCW12*, *PGK1*, *TEF1* and *TDH3* were selected to express *PTS* individually [18], with the yield of patchoulol shown in Figure 2d. Among these promoters, p*GAL1* was identified as the most effective for expressing *PTS*,

resulting in a patchoulol titer of 0.88 ± 0.03 mg/L (0.08 ± 0.001 mg/g DCW), which was 91.3% higher than that expressed by the p*TEF1* strain. Consequently, p*GAL1* was chosen for further experiments involving *PTS* expression.

The *GAL1* promoter is a galactose-induced promoter that is tightly regulated by galactose and glucose. In the existence of glucose, transcription of the *GAL1* promoter is suppressed, requiring the addition of galactose to alleviate this inhibition [19]. However, due to the high cost of galactose and its unsuitability for industrial use, knocking out the transcriptional suppressor *GAL80* switches the regulatory sugars from galactose to glucose. This allows for the activation of exogenous genes controlled by the *GAL1* promoter at low glucose concentrations, removing the dependence on galactose in the GAL induction system [20] (Figure 2e). The recombinant strain Q07, obtained by knocking out *GAL80*, was compared with strain Q02. The results, as depicted in Figure 2f, showed no negative effects on cell growth and yield accumulation upon knocking out *GAL80*. Subsequently, strain TQP01 was created by introducing *PTS* into strain Q07 using a high copy plasmid pESC-URA. Strain TQP01 yielded 2.78 ± 0.19 mg/L (0.26 ± 0.02 mg/g DCW), 3.16 times that of single copy strain Q07 (Figure 2g). Therefore, the engineering transformation will proceed with strain TQP01.

### 3.3. Strengthen the MVA Pathway in Saccharomyces cerevisiae

The MVA pathway in *S. cerevisiae* is an important synthesis pathway of terpenoid compounds, which provides the precursor substance FPP for the synthesis of patchoulol, so it is particularly necessary to strengthen the endogenous MVA synthesis pathway [21]. 3-Hydroxy-3-methylglutaryl-CoA reductase (HMG-CoA synthase, *HMGR*) is a key rate-limiting step in mevalonate synthesis. Previous studies have shown that overexpression of *HMGR* (*tHMG1*) with truncated N-terminal amino acid sequence can enhance the metabolic flow of the endogenous mevalonate pathway in engineered strains and provide a greater flow of five carbon units to the FPP [22,23]. In this study, the *tHMG1* gene was overexpressed in strain TQP01 using a strong promoter, resulting in strain TQP02. After 120 h of fermentation, the yield of patchoulol increased by 4.8 times compared to strain TQP01, reaching 18.95 ± 0.42 mg/L (1.66 ± 0.03 mg/g DCW) (Figure 3a), while squalene yield was 230.68 ± 1.44 mg/L (20.22 ± 0.89 mg/g DCW) (Figure 3b). Subsequent integration of other MVA pathway genes, *ERG20-IDI1*, *ERG19-ERG10-ERG8* and *ERG12-ERG13* genes were divided into three groups (Figure 3c) and progressively integrated into the TQP02 strain to construct the TQP03, TQP04 and TQP05 strains, respectively. After 120 h of shaking, patchoulol yields reached 24.98 ± 2.32 mg/L (2.25 ± 0.25 mg/g DCW), 27.13 ± 0.31 mg/L (2.33 ± 0.0.02 mg/g DCW) and 30.77 ± 0.40 mg/L (2.84 ± 0.12 mg/g DCW), respectively. Strain TQP05 showed a 62.37% increase in patchoulol yield compared to TQP02, but this enhancement was lower than that achieved by overexpressing the rate-limiting enzyme gene (*tHMG1*). To boost the precursor substance FPP of patchoulol accumulation, two additional copies of *tHMG1* were introduced into strain TQP05, resulting in strains TQP06 and TQP07, with TQP07 reaching a patchoulol yield of 48.00 ± 2.46 mg/L (4.36 ± 0.37 mg/g DCW) and a squalene yield of 326.02 ± 0.50 mg/L (29.61 ± 0.98 mg/g DCW) after 120 h of shaker culture.

In summary, by enhancing the expression of endogenous MVA pathway genes in *S. cerevisiae*, patchoulol yield increased significantly by 16.2 times (from 2.78 mg/L to 48.00 mg/L), and squalene production, utilizing FPP as a precursor, also increased from 0.48 mg/L to 326.02 mg/L (Figure 3b). These modifications did not negatively impact strain growth, suggesting that enhancing the metabolic flux of acetyl-CoA to FPP in the MVA pathway is beneficial for increasing patchoulol production.

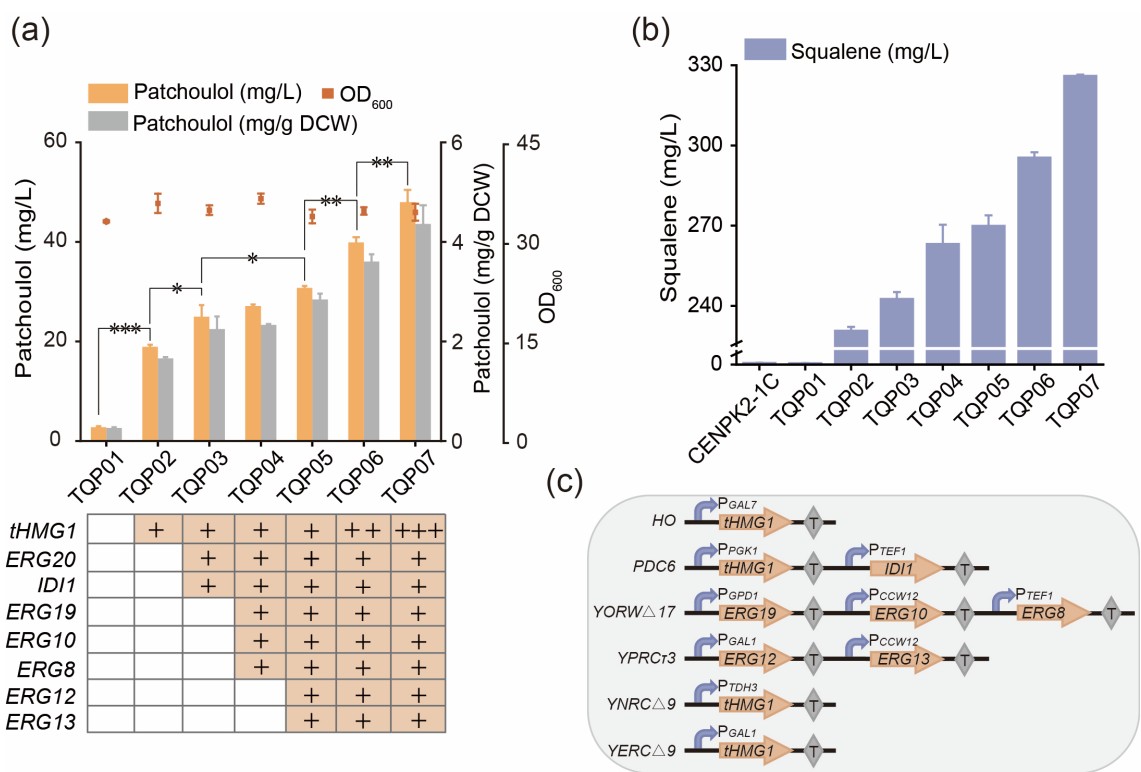

**Figure 3.** Strengthen the MVA pathway in *S. cerevisiae*. (**a**) Effects of overexpression of mevalonate pathway genes on the yield and cell growth of patchoulol; (**b**) Effects of overexpression of mevalonate pathway genes on squalene in *S. cerevisiae*; (**c**) Construction of gene expression frame of the mevalonate pathway. Data presented as mean values ± SD from three independent biological replicates (*n* = 3). Statistical evaluation (*p*-value) compared to the control was conducted by a two-tailed *t*-test. * $p < 0.05$, ** $p < 0.01$, *** $p < 0.001$ ($p \geq 0.05$).

### 3.4. Reduce the Consumption of Precursor FPP by Balancing Competition Pathway

FPP plays a crucial role in the isoprene pathway as an intermediate, serving as a precursor for various metabolites. The farnesol pathway and squalene pathway, which use FPP as a precursor, compete with the patchoulol synthesis pathway for FPP and affect patchoulol synthesis [24,25]. Inhibiting the competition pathway of FPP is an important strategy to enhance patchoulol production. Previous research has indicated that the transformation of FPP to farnesol is primarily regulated by *DPP1* and *LPP1* genes, with their absence leading to increased sesquiterpene yields [26]. By sequentially knocking out *DPP1* and *LPP1* genes in the TQP07 strain, new strains TQP08, TQP09 and TQP10 were constructed. Experimental results (Figure 4a,b) showed that knocking out *DPP1* and *LPP1* genes boosted patchoulol production, with TQP10 achieving a yield of 60.94 ± 0.46 mg/L (5.77 ± 0.03 mg/g DCW), a 26% improvement over TQP07. However, the impact of the knockout of the farnesol pathway was not significant, especially in the strain with the *DPP1* knockout alone. This is similar to the results obtained by Liu et al. [6] This is likely due to the proteins regulated by the *DPP1* and *LPP1* genes being small molecular weight integrated membrane proteins located on the Golgi apparatus and vacuole membranes, while the synthesis of patchoulol is mainly in the cytoplasm. There is a limited conversion of FPP to farnesol in the cytoplasm via *DPP1* and *LPP1*, with more FPP flowing to other substances such as squalene, heme A and ubiquinone [24]. Therefore, reducing the metabolic flux of farnesol is limited to increasing the production of zebra mussels.

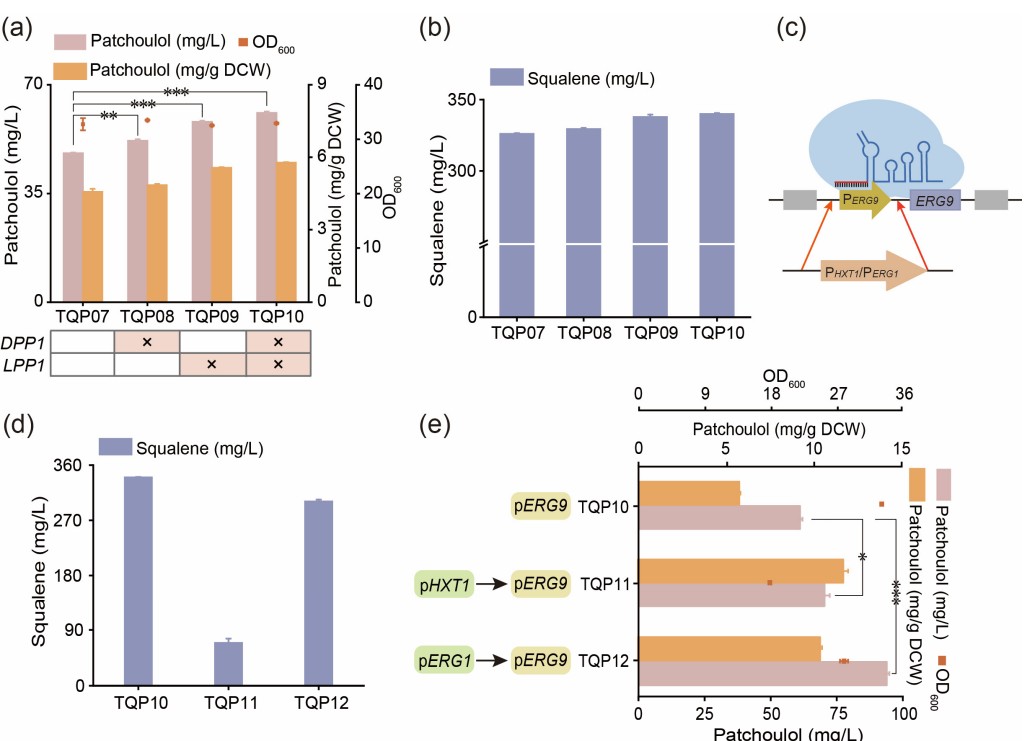

**Figure 4.** Reducing the consumption of precursor FPP by balancing the competitive pathway of patchoulol. (**a**) Effects of knockout of farnesol pathway genes *DPP1* and *LPP1* on production and growth; (**b**) The effect of knockout of farnesol pathway genes *DPP1* and *LPP1* on squalene; (**c**) Using CRISPR/Cas9 gene editing technology to replace the promoter of squalene synthase; (**d**) The effect of weakening squalene pathway on squalene in *S. cerevisiae*; and (**e**) Effects of weakening squalene pathway on the yield and cell growth of patchoulol. Data presented as mean values $\pm$ SD from three independent biological replicates ($n = 3$). Statistical evaluation (*p*-value) compared to the control was conducted by a two-tailed *t*-test. * $p < 0.05$, ** $p < 0.01$, *** $p < 0.001$ ($p \geq 0.05$).

In *S. cerevisiae*, FPP is turned into squalene through squalene synthetase, which is further transformed into ergosterol through a series of enzymatic reactions. Since ergosterol plays a vital role in the cell membrane composition and *S. cerevisiae* is unable to uptake exogenous ergosterol in aerobic conditions, direct knockout of the gene is not feasible [8]. A common approach to address substrate competition issues involves regulating the expression of *ERG9* through its promoter to modulate squalene metabolism (Figure 4c). To dynamically control *ERG9* expression, we opted for the *HXT1* promoter, which responds to changes in glucose levels [27]. By replacing the native *ERG9* promoter in strain TQP10 with the *HXT1* promoter to generate strain TQP11 (as depicted in Figure 4d,e), the production of patchoulol increased to 70.13 $\pm$ 1.95 mg/L (11.60 $\pm$ 0.21 mg/g DCW), while squalene content decreased significantly from 340.00 mg/L to 69.03 mg/L. However, this led to a 46.2% reduction in yeast biomass due to the decreased ergosterol flux, impacting cell growth. To mitigate this, we replaced the p*ERG9* promoter with *ERG1*, which is regulated by ergosterol levels, in strain TQP12. This adjustment resulted in a higher patchoulol yield of 93.79 $\pm$ 0.96 mg/L (10.33 $\pm$ 0.08 mg/g DCW), a decrease in squalene content from 340.00 mg/L to 300.65 mg/L and a reduction in OD$_{600}$ from 32.92 to 27.94 (Figure 4e). Notably, p*ERG1* had a milder effect on cell growth compared to p*HXT1*, making it a better choice for balancing the squalene competition pathway.

In conclusion, knocking out the *DPP1* and *LPP1* genes, which regulate farnesol, led to a modest increase in patchoulol yield. Dynamic regulation of the *ERG9* gene expression through the ergosterol-responsive promoter *ERG1* enabled cells to adjust the metabolic flux of the squalene synthesis module in accordance with ergosterol levels, facilitating the production of patchoulol [7].

### 3.5. Improving the Production of Patchoulol by Modifying Transcription Regulatory Factors

Modifying transcription regulatory factors in chassis cells has been shown to increase the yield of terpenoids in *S. cerevisiae* [28]. This study explored common transcriptional regulatory factors to increase the production of patchoulol. Knocking out *ROX1*, a transcriptional suppressor of hypoxic genes, was found to boost the transcription and translation of MVA pathway genes, thereby intensifying the mevalonate pathway [29]. Deficiency in *YPL062W*, *YJL064W* and *YNR063W* (transcription regulatory factors, their genetic functions and molecular mechanisms are currently unclear) yeast strains was observed to alter the carbon flow of acetyl-CoA, enhance glucose utilization, and improve energy metabolism [6,30,31]. The mutant gene *UPC2-1*, derived from *UPC2*, was found to enhance the absorption of exogenous sterols in cells under aerobic conditions and elevate the expression level of the sterol metabolic pathway [32]. In this study, *YPL062W*, *YNR063W*, *YJL064W* and *ROX1* genes were individually or jointly knocked out in TQP12 strain, and the *UPC2-1* gene was overexpressed, resulting in the creation of engineered strains TQP13, TQP14, TQP15, TQP16, TQP17, TQP18 and TQP19. The production of patchoulol in these strains was measured as $98.13 \pm 2.57$ mg/L, $96.42 \pm 2.92$ mg/L, $69.03 \pm 1.31$ mg/L, $103.34 \pm 1.54$ mg/L, $110.14 \pm 3.05$ mg/L, $65.83 \pm 2.03$ mg/L and $53.40 \pm 3.77$ mg/L, respectively (Figure 5a). The results indicated that knocking out *YPL062W*, *YNR063W* and *YJL064W* individually showed some increase in patchoulol accumulation, the combined deletion of these three genes significantly boosted patchoulol yield, suggesting a synergistic effect of these gene deletions on patchoulol accumulation.

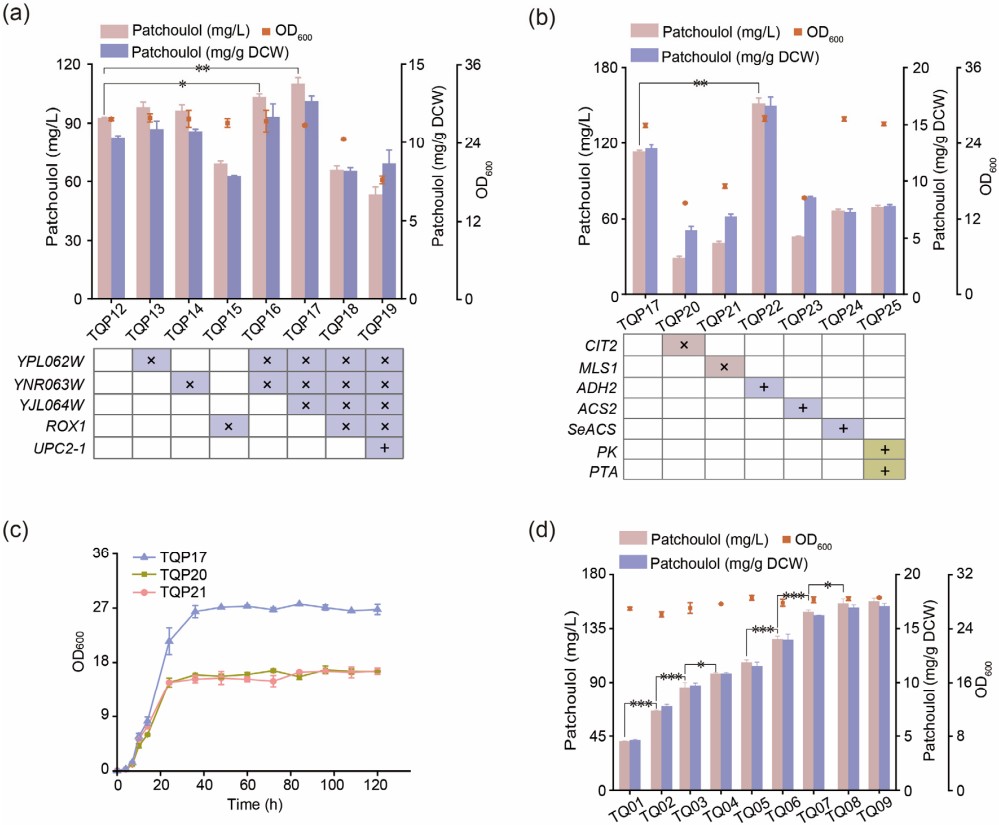

**Figure 5.** Effects of modified transcriptional regulators and regulation of acetyl CoA on cell growth and patchoulol production. (**a**) The effect of modified transcriptional regulators on yield and growth; (**b**) The influence of regulating the acetyl-CoA pathway on yield and growth; (**c**) The absence of acetyl-CoA competitive pathway genes *CIT2* and *MLS1* led to growth arrest in *S. cerevisiae*; and (**d**) Effects of *PTS* multi-copy integration on the production of patchoulol. Data presented as mean values $\pm$ SD from three independent biological replicates ($n = 3$). Statistical evaluation ($p$-value) compared to the control was conducted by a two-tailed $t$-test. * $p < 0.05$, ** $p < 0.01$, *** $p < 0.001$ ($p \geq 0.05$).

Unexpectedly, the knockout of the *ROX1* gene and overexpression of the *UPC2-1* gene led to a decrease in patchoulol production. Jordá et al. [33] observed increased mRNA and protein levels of the *ERG1* (squalene epoxygenase-encoding gene) gene after knocking down the *ROX1* gene. We guessed that an enhancement in the metabolic pathway of epoxidized squalene may have competed with the patchoulol pathway for the precursor FPP, resulting in reduced patchoulol production. Leak et al. [34] noted that the *UPC2-1* mutant strain displayed heightened sensitivity to metal cations and increased cell membrane permeability, which possibly makes a defective cytoplasmic membrane. Therefore, the decrease in patchoulol yield following the overexpression of the *UPC2-1* gene could be attributed to cytoplasmic membrane defects in the recombinant strain, affecting yeast growth.

### 3.6. Regulating the Acetyl-CoA Pathway to Promote the Synthesis of Patchoulol

In yeast, the MVA pathway starts with acetyl-CoA as a precursor for synthesizing terpenoids, and insufficient acetyl-CoA supply can impact terpenoid biosynthesis in engineered *S. cerevisiae* [35]. Therefore, increasing the content of acetyl-CoA in *S. cerevisiae* could have a positive significance in patchoulol synthesis. This study primarily focused on three aspects: a. Reducing the consumption of cytoplasmic acetyl-CoA. The glyoxylate cycle in *S. cerevisiae* involves *CIT2* and *MLS1*, which compete with the MVA pathway for acetyl-CoA [35]. To investigate this, the two genes were knocked out in the TQP17 strain to create TQP20 and TQP21, respectively. The results revealed a significant decrease in patchoulol production in strains TQP20 and TQP21 (Figure 5b), likely due to disruption of the glyoxylate cycle, impacting *S. cerevisiae*'s metabolic pathway and leading to growth arrest (Figure 5c). b. Strengthening the endogenous acetyl-CoA metabolic pathway in the cytoplasm of *S. cerevisiae*. During *S. cerevisiae* growth, a substantial amount of ethanol is produced, which can be converted to acetaldehyde by the *ADH2* gene. However, the natural promoter of *ADH2* is repressed in the presence of glucose [36]. In this study, a strong component promoter was utilized to overexpress the *ADH2* and *ACS2* genes of acetyl-CoA synthase in the TQP17 strain, resulting in the creation of TQP22 and TQP23 strains, respectively. Strain TQP22 exhibited a patchoulol yield of 151.27 ± 4.77 mg/L (16.66 ± 0.76 mg/g DCW), 1.34 times higher than that of TQP17, while the growth and yield of strain TQP23 decreased significantly (Figure 5b). According to the literature, the efficiency of the *ACS* variant $SeACS^{L}641^{P}$ from *Salmonella enterica* was higher than that of endogenous *ACS2* in *S. cerevisiae* [37]. $SeACS^{L}641^{P}$ was overexpressed on strain TQP17, but the production of patchoulol still decreased. This outcome may be attributed to the regulation of acetyl-CoA synthase by *S. cerevisiae* and the enzyme's activity. Increased expression of acetyl-CoA synthetase may lead to the excessive accumulation of toxic precursor acetyl-CoA in the strain, disrupting acetyl-CoA balance and impacting strain growth. c. Constructing a heterologous acetyl-CoA synthesis pathway. Under the catalysis of phosphotransferase (*PK*) and phosphotransferase (*PTA*), xylose 5-phosphate can be converted to acetyl-CoA without releasing carbon dioxide [38]. In this research, *PK* from *Bifidobacterium longum* and *PTA* from *Clostridium kluyveri* were co-expressed in *S. cerevisiae*. Consequently, the production of patchoulol decreased significantly. This outcome may be attributed to the fact that this pathway does not yield ATP and NADPH, suggesting that coupling with another pathway capable of producing NADPH may be necessary to enhance acetyl-CoA levels.

Considering the lack of genetic stability of strains overexpressing *PTS* using multicopy plasmids during long-term passage, the pESC-PTS plasmid was eliminated from the TQP22 strain using SD/- Trp+5-FOA solid culture medium to generate strain Q22. *PTS* was then gradually integrated into the chromosome of strain Q22, resulting in strains TQ01-TQ09. As shown in Figure 5d, the yield of patchoulol increased with the number of *PTS* copies integrated. When eight copies of *PTS* were integrated, the yield reached 155.94 ± 3.76 mg/L, similar to the plasmid-expressing strain. Further increases in *PTS* copies did not lead to higher yields. Therefore, strain TQ08 was selected for the next experiment.

### 3.7. Optimize the Fermentation Process to Improve the Production of Patchoulol

The growth of *S. cerevisiae* requires appropriate culture conditions. In order to optimize the synthesis of patchoulol from recombinant *S. cerevisiae* strain TQ08, the culture medium was adjusted based on YPD. The optimized culture conditions included 30 g/L glucose, 30 g/L yeast extract, 20 g/L tryptone, 1.232 g/L $MgSO_4 \cdot 7H_2O$, 4.8 g/L $KH_2PO_4$, 10 mL/L metal ion solution A and 24 mL/L vitamin solutions B. Fermentation of strain TQ08 in shaking flasks using the optimized medium (OYPD) resulted in a yield of 195.96 ± 1.61 mg/L (16.33 ± 0.32 mg/g DCW), which was 30.74% higher than YPD.

To enhance patchoulol yields, strain TQ08 was fermented in a 5-L bioreactor. Given that strain TQ08 is a trophic mutant strain, the medium was supplemented with 1.0 g/L leucine, 1.0 g/L histidine and 1.0 g/L uracil to promote optimal growth of the strain.

The study utilized a two-stage fed-batch fermentation process with optimized conditions using 10% dodecane for the extraction of patchoulol (Figure 6b). In the first stage, the engineered strain consumed nutrients in the culture medium for 18 h until glucose depletion, followed by the addition of feed medium I to promote rapid cell growth. Glucose concentration was carefully controlled below 1.0 g/L by adjusting the flow rate of feed medium I. Ethanol is commonly used as a carbon source in terpenoid-producing *S. cerevisiae* during fed-batch fermentation and greatly increases terpenoid yields compared to glucose [23,39]. In the second stage, after 48 h, strain TQ08 biomass remained stable with an $OD_{600}$ of 185.31 and ethanol content of 31.0 g/L. Subsequently, feed medium I was replaced with feed medium II to supplement the carbon source, with the ethanol flow rate adjusted to maintain a concentration of about 20 g/L. As shown in Figure 6b, the patchoulol yield gradually increased throughout the fermentation process. In the first stage, the cell concentration of the strain increased sharply, but the patchoulol yield increased slowly, reaching only 246.42 mg/L at 48 h. In the second stage, the strain stopped growing and the flow-added carbon source was shifted from glucose to ethanol to promote the synthesis of patchoulol, which was observed to increase significantly, reaching 1.95 g/L at 168 h.

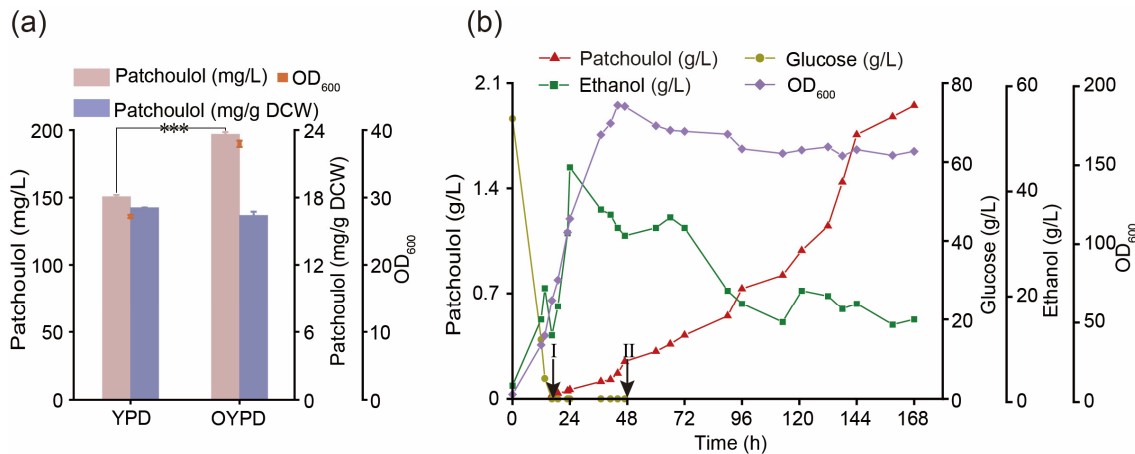

**Figure 6.** Optimizing fermentation conditions to improve the production of patchoulol. (**a**) Effects of YPD and OYPD media on the yield and growth of strain TQ08; (**b**) Overproduction of patchoulol by TQ08 strain in batch feed fermentation. Data presented as mean values ± SD from three independent biological replicates (*n* = 3). Statistical evaluation (*p*-value) compared to the control was conducted by a two-tailed *t*-test. *** $p < 0.001$ ($p \geq 0.05$).

### 4. Conclusions

In this study, a strain of *S. cerevisiae* was engineered to produce high levels of patchoulol using a combinational metabolic engineering strategy. By optimizing the patchoulol metabolic network, including screening the best promoter for *PTS*, strengthening the mevalonate pathway, balancing competing metabolic pathways, modifying transcriptional regulators and regulating the acetyl-CoA pathway, the patchoulol yield of *S. cerevisiae*

was significantly increased. Further optimization of fermentation medium and fed-batch fermentation techniques ultimately resulted in a patchoulol yield of 195.96 mg/L in shake flasks and 1.95 g/L in a 5-L bioreactor. Among the strategies employed, overexpression of the *tHMG1* gene, a key enzyme gene in the MVA synthesis pathway, along with downregulation of the *ERG9* gene through the *ERG1* promoter, and overexpression of the *ADH2* gene using a strong promoter were key factors contributing to the increase in patchoulol yield, and these strategies provided more precursors for the synthesis of patchoulol, which resulted in a significant increase in the patchoulol yield. This study shows that an insufficient supply of precursors is one of the main factors limiting the synthesis of patchoulol in our background strain. Strengthening the precursor supply positively impacts patchoulol synthesis, crucial for *S. cerevisiae*. This study also holds relevance for other sesquiterpenoids.

**Supplementary Materials:** The following supporting information can be downloaded at: https://www.mdpi.com/article/10.3390/fermentation10040211/s1, Figure S1: The relationship between cell dry weight and $OD_{600}$, y is the cell concentration, unit g DCW/L, x is $OD_{600}$; Figure S2: Schematic diagram of strain construction. "Δ" indicates knocking out; "Δ*A:: B*" indicates overexpression of B gene at site A; Table S1: Primers Used in This Study; Table S2: Exogenous gene sequences used in this study.

**Author Contributions:** Conceptualization, Q.T. and Z.P.; Methodology, Q.T. and Z.P.; Validation, Q.T. and Z.P.; Writing—original draft, Q.T. and Z.P.; Writing—review & editing, Q.T., J.Z. and Z.P.; Supervision, G.D., J.C., J.Z. and Z.P. All authors have read and agreed to the published version of the manuscript.

**Funding:** This research received no external funding.

**Data Availability Statement:** Data are contained within the article and Supplementary Materials.

**Conflicts of Interest:** The authors declare no conflict of interest.

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
