# Peer review of "Metabolic Engineering for Efficient Synthesis of Patchoulol in Saccharomyces cerevisiae"

_fermentation, doi:10.3390/fermentation10040211_

Round 1

Reviewer 1 Report

Comments and Suggestions for Authors

In this study, the authors reported the efficient synthesis of patchoulol in Saccharomyces cerevisiae using a combinational metabolic engineering strategy. The production of patchoulol in S. cerevisiae was increased by screening the optimal promoter of allozyme, enhancing mevalonate pathway, balancing competitive metabolic pathway, modifying transcriptional regulatory factors and regulating acetyl-CoA pathway The yield was further improved through fermentation optimization and fed-batch fermentation I think this is an important study and justifies the publication of this manuscript in Fermentation once the following issues are addressed.

1)     Patchoulol is used in the synthesis of the chemotherapy drug Taxol. It’s important to mention that in the introduction.

2)     Figure 1. Citrate instead of citrin

3)     Figure 2. (b) Please specify whether it’s an LCMS or GCMS chromatogram. There is an additional peak apart from patchoulol. Please reveal the identity of this additional peak.

4)     Throughout the manuscript, the authors used multiple y-axis to show different parameters. Please change the figures.

5)     Figure 4. (a) and (e). It’s very hard to see the OD600 values. Please change the figures accordingly.

6)     Figure 6. (a). OD600 values are not visible.

Reviewer 2 Report

Comments and Suggestions for Authors

The manuscript “Metabolic Engineering for Efficient Synthesis of Patchoulol in Saccharomyces cerevisiae” by Tao et al., reports the recombinant production of patchoulol, a natural sesquiterpene with biotechnological applications, in yeast. Authors performed several genetic strategies including promoter analysis for patchoulol synthase (PTS) expression, integration of the expression cassette into the genome or multicopy plasmids, knocking down competitive pathway genes among others, and optimizing fermentation conditions to enhance metabolite production. Scientifically, the manuscript is interesting and aligns with the scope of the journal. However, statistical analyses are recommended to support the significance of the differences that were observed. In terms of the text, it requires improvement for clarity and precision, and assistance for the English language is required. A comprehensive review by the authors is recommended. I hope my feedback will help the authors to improve the manuscript.

The text occasionally is unclear. For example, the introduction mentions the insertion of a heterologous synthesis pathway for patchoulol production (L44). However, only one enzyme (the patchoulol synthase, PTS) is required for its synthesis in yeast, which is a detail not adequately explained in the text.

L45 indicates "....the metabolic flow of squalene synthesized by farnesyl pyrophosphate (FPP)...." which is scientifically inaccurate. FPP does not synthesize squalene, it serves as a substrate for the enzyme encoded by the ERG9 gene. Similar to this example, phrases along the text lacking precision when using scientific terminology are included and the text should be carefully revised.

Methodological descriptions have inconsistent verb tenses. Sometimes it explains how the experiments were done, but other paragraphs include "instructions" (like a protocol; for example, see L134-L137). Some phrases are not clear. For example, L146 indicates "the supernatant was poured out and mixed with grinding beads (0.5 mm) of the same volume as the cell precipitation, and then 1.5 mL acetone was added": as is, I understand that the supernatant was mixed with the beads, but it should be the cell pellet.

The text in the description of the results requires frequent review of previous sections for understanding, and there are "explanatory gaps" that require the reader to “guess” to understand the experimental design rationale. For example, it should be indicated when PTS gene was integrated into the genome or in plasmid.

Strain Q07 was compared with Q02 (Fig. 2). From table 2 it is understood that Q07 derives from Q06, which had the TDH3 promoter, but strain Q02 had the GAL1 promoter to express PTS, therefore, I do not think that this comparison (Q02 vs Q07) is appropriate.

Strain Q07 is transformed with a multicopy plasmid with the PTS expression cassette with the Gal1 promoter, obtaining strain TQP01. So, does this last strain have both the PTS gene integrated into the genome (as it derives from Q07) and on the multicopy plasmid?

I strongly suggest including a schematic representation that summarizes the flow of how strains were obtained, highlighting the new modifications in each resulting strain. This will greatly help the reader to follow the manuscript.

L242: it would be good to indicate which genes were overexpressed in each strain.

L276-L278: it is not clear what the authors intended to indicate.

L289-290: indicates that “FPP is turned into squalene through squalene synthetase, which is 289 then catalyzed to ergosterol by a series of enzymes”, however, Fig1 only shows a single arrow with the ERG1 gene from squalene to ergosterol, which is confusing for a reader not familiar with ergosterol synthesis.

L292: From squalene to ergosterol there are several steps catalyzed by different enzymes. Which gene are the authors referring to?

In several graphs in the figures, the OD600 value is included, but the legend does not give indications. Is this the OD of the culture when the sample was taken? Was the culture at the stationary phase? etc. The OD information is included in the figures, but in several parts of the text, this is not clarified or discussed, which should be particularly important when discussing the growth alterations that resulted from some of the genetic modifications that were done. The OD600 value is not visible in Fig. 4e, which is important since the text mentions that the modification made in strains affected growth.

L326: what are YPL062W, YJL064W, and YNR063W?

The OD600s are not visible in Fig. 5a.

L336: Why was the physiological state of yeast cells improved? What result suggests this? How was this evaluated? Are the changes in patchoulol production significant? This should be discussed.

L349-L352: How did the authors evaluate what these lines indicate? In the way this is presented, it seems more like a fact, but this was not evaluated (there are no gene expression data in the manuscript).

L357: rather than "has a very positive," it should say "could have a positive," and therefore the authors made the following modifications. Please replace "definitive statements" with "conditional ones", when corresponding.

L369-370: ADH2 repression by ethanol. Please verify this statement and include reference support.

L378-379: this discussion requires a better elaboration.

L387-389: It is not clear how the transformation of each strain was carried out (by integration or plasmid). Therefore, as mentioned before, a detailed schematic flow representation of how the strains were obtained is recommended to help the reader. Considering the information in Table 2, it is understood that the TQP strains derive from Q07, which in turn derive from Q06. From Figure 2a, the reader can interpret that strains Q01-07 were obtained by integration and that the first TQP strain (01) was obtained by transforming Q07 with pESC-PTS. Therefore, the TQP strains contain the PTS cassette integrated (as in strain Q06) and on a multicopy plasmid (as in strain Q07, which in turn derives from Q06). Strain Q22 is not described in the table, so it is unclear how it was obtained or its characteristics.

L391: The method by which eight copies of the PTS gene were integrated is unclear and lacks verification.

L397-404: What criteria were used to design the composition of the OYPD medium to favor the metabolite production?

L410-411: Why add supplements if OYPD medium is used? And why tryptophan was not considered?

L419-420: the OD value seems very high to me, and the numbers coincide exactly with the value of the yield of patchouli alcohol (185.31 mg/L) mentioned in L424. Please check.

L420: The phrase: "Firstly, stop supplementing the carbon source to reduce ethanol to below 20 g/L.", is unclear and not well connected with the text.

Fig6b: incorporating “arrows” indicating when glucose and ethanol were added could benefit the figure to facilitate its understanding.

A comprehensive discussion of the fermentation results is missing.

The conclusions are very general, need to be more precise, and based on the results, specifically regarding the key factors that contributed (and which of them did not) to enhance the production of patchoulol in yeast.

Comments on the Quality of English Language

The text requires substantial improvement in precision and English language.

Round 2

Reviewer 2 Report

Comments and Suggestions for Authors

The manuscript “Metabolic Engineering for Efficient Synthesis of Patchoulol in Saccharomyces cerevisiae” by Tao et al., reports the recombinant production of patchoulol, a biotechnologically significant natural sesquiterpene alcohol, in yeast.

The revised version largely improved regarding my previous comments and in the English language. Still, I have some comments:

Previous comments:

1) L369-370: ADH2 repression by ethanol. Please verify this statement and include reference support.

Author Response: Thank you for your suggestion. We have cited an article in this study to support this statement. (Lines 404).

Response: The cited article refers to repression by glucose, which is well-known for the ADH2 gene. However, my concern was regarding if ethanol represses this gene as these lines say (ethanol is the substrate of the enzyme encoded by ADH2). Please check.

2) L419-420: the OD value seems very high to me, and the numbers coincide exactly with the value of the yield of patchouli alcohol (185.31 mg/L) mentioned in L424. Please check.
Author Response: Thank you for your suggestion. We are very sorry for our negligence in recording the production of patchoulol in the first stage incorrectly. The production of patchoulol in the first stage was 246.42 mg/L, and we have corrected this data in this study. (Lines 467). Perhaps due to the different construction processes and batch feeding methods of the strains, after many experiments, OD600 of the final strain constructed in this paper was between 160-195 in a 5-L fermenter.

Response: Thank you for your response. However, the OD values still seem very high and not biologically possible. How were ODs calculated? Were OD values measured directly by diluting the culture or were estimated from the DCW/L using the calibration curve shown in Fig. S1? If it is this last case, I do not think the curve is appropriate as the OD range goes from 0 to 25 and the reported value is way above this. 

Please check if the OD values in the whole manuscript were well estimated. Usually, a yeast culture reaches an OD between 10-15 at the stationary phase when cultured in flasks. Probably higher OD values can be reached by yeast cultures in fermentors, but the reported values seem very high. Please check and explain.

3) The conclusions are very general, need to be more precise, and based on the results, specifically regarding the key factors that contributed (and which of them did not) to enhance the production of patchoulol in yeast.

Author Response: Thank you for your suggestion. We have reorganized the conclusions of this paper based on our experimental results. And we hope this is appropriate. (Lines 472-494)

Response: Thank you for considering my comment. However, the new conclusion is only a summary of your results (not a conclusion). Please refer to the main findings of your work and key factors that contributed to enhancing the production of the metabolite in yeast. 

New comments:

4) L334-337: Please correct and improve this paragraph. For example, ergosterol does not have a promoter (as the text says). In addition, is not possible to conclude that the modifications that were made maintained intracellular sterol levels at an appropriate range, as sterols were not measured (the work evaluated growth).

5) L378: “….bacterial growth….”, should say “yeast growth”.

Comments on the Quality of English Language

The English language is understandable, but it can be improved.
